# The experience of noise in communication-intense workplaces: A qualitative study

**Kristina Gyllensten**[1]*, **Sofie Fredriksson**[2], **Stephen Widen**[3], **Kerstin Persson Waye**[2]

**1** Department of Occupational and Environmental Medicine, University of Gothenburg, and Sahlgrenska University Hospital, Gothenburg, Sweden, **2** Occupational and Environmental Medicine, School of Public Health and Community Medicine, Institute of Medicine, University of Gothenburg, Gothenburg, Sweden, **3** School of Health and Medical Sciences, Örebro University, Örebro, Sweden

* kristina.gyllensten@amm.gu.se

## Abstract

### Objective

The aim of the study was to explore and describe how workers in communication-intense workplaces in health care and preschools experience the sound environment. The dependence on vocal communication and social interaction poses a challenge using hearing protection in these working environments.

### Method

A qualitative method was used, more specifically inductive thematic analysis was used, as this approach was deemed suitable to explore the staff's experiences of the sound environment. Data were collected by interviews and to increase trustworthiness, several researchers were involved in the data collection and analysis.

### Study sample

Workers from two preschools, one obstetrics ward and one intensive care unit took part in the study.

### Results

Four main themes emerged from the thematic analysis: A challenging and harmful sound environment; Health-related effects of a challenging and harmful sound environment; A good sound environment is not prioritised; and Resourceful and motivated staff.

### Conclusions

Workers in communication-intense workplaces in preschools, obstetrics care and intensive care reported that there was a relationship between the sound environment and negative health effects. In addition, the results suggests that the high motivation for change among staff should be utilised together with an increased prioritization from the management to reach innovative context specific improvements to the sound environment in communication intense working environments.

**Data Availability Statement:** Data cannot be shared publicly because potentially attributable sensitive information about health and symptoms regarding the participants. And when sharing such

data there has to be an approval from a Swedish Ethical committee. However, anonymised data is available with an approval from an ethical review board. For data requests, contact: Department of occupational and environmental medicine, Gothenburg University, amm@amm.gu.se or Kristina Gyllensten, Department of occupational and environmental medicine, Gothenburg University, kristina.gyllensten@amm.gu.se. The name of the data set is 'FriskArb: Noise in female-dominated occupations, the qualitative study.

**Funding:** - The study was funded by Forte - Grant number: 2016-07193 - KPW, SF, SW and KG received the award funded by Forte - https://forte.se/ - The funders had no role in study design, data collection and analysis, decision to publish, or preparation of the manuscript.

**Competing interests:** The authors have declared that no competing interests exist.

## Introduction

Noise is commonly defined as undesirable or unwanted sounds that have a negative impact on an individual's physiological or psychological wellbeing [1]. This definition may be challenged as also wanted sounds of high noise levels may be harmful to health. In this paper we therefore adopt the definition of noise as "any sounds at hazardous levels or sounds that are perceived as unwanted". In addition, we define the more specific concept of communication-intense noise as noise that consists of pervasive speech communication or important and meaningful acoustic information.

Occupational noise exposure is one of the most prominent occupational hazards worldwide [2]. The detrimental health effects can be auditory, such as hearing loss, tinnitus or hyperacusis, or non-auditory, such as annoyance, reduced cognitive performance, or stress arousal [2–5]. Sound level and exposure time are generally considered the main predictors of auditory effects, particularly hearing loss [6].

In a preschool and hospital work environment the main type of noise is communication-intense noise. In preschools, children talking or screaming, their activities, and intense conversation in general have been reported in surveys as the most annoying or disturbing sounds for the personnel [7]. In a large survey including almost 5,000 preschool teachers, as many as 70% reported noise annoyance at work [8] In addition, numerous studies conducted in preschools have measured high and potentially hearing-damaging sound levels of around 80 decibel A-weighted equivalent levels (dBA Leq) in personnel dosimetry [9], with repeated intermittent levels above 85 dBA Leq assessed in 1-second loggings [7]. In obstetrics care, equivalent sound levels have been found to reach or exceed 80 dBA in almost half (46%) of the work shifts and 85 dBA in 5% of the shifts [10]. In the same study, almost half of the surveyed staff (49%) reported noise annoyance. One of the few studies evaluating the sound environment in the obstetrics, identified that the sources of loud noise in delivery rooms, defined as sound levels above 90 dB sound pressure level, SPL, were mothers and newborns screaming loudly and many people talking at the same time [11]. In the intensive care unit (ICU), a survey found that 44% of staff reported annoyance, with the main source being alarms from medical equipment, but also conversations between personnel [12]. Equivalent sound levels in the ICU were around 50–60 dBA in patients' rooms, and close to 70 dBA-equivalent levels in personnel dosimetry [13].

In contrast to surveys and measurements, there is a significant lack of knowledge from qualitative studies regarding personnel experience of noise in communication-intense work environments. Qualitative studies provide the possibility to gain more in depth knowledge of how the respondents add meaning to, relate to or cope with agents in the environment. A recent qualitative study using thematic analysis with staff working in an ICU in Turkey [14] found that "human-induced noise" was perceived to have a negative effect on work performance by disturbing concentration, having a negative effect on decision making and making it more likely to make mistakes. Moreover, the study found that if preventive measures were not systematically implemented, their effect was merely short lasting/momentary. To prevent negative health effects of noise exposure, the goal is typically to reduce the noise level at the source. However, in communication-intense sound environments, the main source of the noise is human interaction, speech communication or acoustic alarm signals, all of which are central to the working activities. These sources cannot easily be attenuated and use of hearing protection devices could cause communication difficulties. Wearing hearing protection has been reported by preschool teachers as unpleasant in front of parents, and as hindering the fulfilment of teaching duties [15]. Only a few intervention studies have been performed in these types of workplaces, often resulting in rather modest noise reduction [9, 16]. The impacts on

the personnel and children have though been slightly positive. As for intervention studies within health care, most studies relate to intensive care units (ICU) with none within the obstetrics. Some report significant noise reduction whilst others have not as reported in the review by [17]. Apart from a conference paper indicating that personnel perceived the environment in an ICU to be less noisy after an acoustic and visual intervention [18], there is little guidance of the personnel response.

There is hence a lack of knowledge about what the personnel in communication-intense sound environments experience to be the specific problems and specific needs relating to their sound environment, as well as about the feasibility of implementing preventive measures in a complex interactive workplace.

## Aim

The aim of the present study was to describe how workers in communication-intense workplaces, in health care and preschools, experience their sound environment. In addition, the purpose was to identify factors of importance for a subsequent intervention study, which will focus on improving the work environment and in particular the sound environment in these workplaces.

## Materials and methods

### Participants

A purposeful sampling strategy was used to ensure that the sample consisted of participants with relevant experience relating to communication-intense workplaces and noise at work in health care and preschool. In total, 16 individuals participated, including staff and managers from four selected workplaces. The workplaces included two preschools under the same manager, one delivery department in an obstetrics ward, and one post-operative intensive care department in a hospital. The two preschools were located in a small city in Sweden, each with 20 employees. The obstetrics ward was located at a hospital in a large city in Sweden and had approximately 100 full or part-time employees. The ICU was located at a hospital in a medium to large city in Sweden with approximately 70 full or part-time employees. The workplaces were initially approached by the researchers in 2016, on the basis that they were communication-intense workplaces located in the same region as the research group. The obstetrics ward was approached directly after having taken part in another noise-related study [10]. The ICU ward was approached indirectly via a staff member from another ICU ward that had taken part in a previous noise-related study [19]. The preschools were indirectly recruited via their municipal headmaster, who was approached by the researchers about whether any preschools in the municipality would be interested in participating. The managers at the participating workplaces invited all relevant staff to participate in the study, and all employees who volunteered were included. They received no reimbursement for their participation. Participants are described in Table 1.

### Procedure and data collection

The data were mainly collected using semi-structured focus group interviews. This approach allows for flexibility during the interviews and a sharing, collaborative discussion of different experiences and opinions among and interaction between participants [20]. The managers were interviewed individually to ensure that participants could answer freely without their superiors present. The interview guide, which had been developed in relation to the research aim and existing literature, contained open questions about the sound environment and the

**Table 1. Information regarding participants and focus groups.**

| | Obstetrics ward | Intensive care department | Preschools |
|---|---|---|---|
| Occupation | 3 midwives | 3 nurses | 6 preschool teachers |
| | 1 manager | 2 managers | 1 manager |
| Gender | 4 females | 5 females | 7 females |
| Age range | 45–66 years | 51–65 years | 41–49 years |
| Number of participants in each interview | One focus group interview with 3 midwives | One focus group interview with 3 nurses | Two focus group interviews with 3 preschool teachers in each |
| | One individual interview with one manager | One interview with 2 managers | One individual interview with one manager |
| Interviewers | First interview: KG[1] and SW[2] | First interview: KG[1] and SW[2] | First interview: KG[1] and SW[2] |
| | Second interview: KG[1] | Second interview: KG[1] | Second interview: KG[1] and SW[2] |
| | | | Third interview: KG[1] |

1 = Fist name Surname

2 = First name Surname

psychosocial working environment. Examples of questions included the following "Can you describe the sound environment at the workplace?" "How do you handle disturbing noise at the workplace?", "What changes could be made to create a better sound environment at the workplace?" The interview guide did not differ between staff and managers. As described in Table 1, in total seven interviews were conducted. All interviews were conducted at the participants' workplaces. Participants in the focus groups knew each other. The interviewers therefore paid close attention to the group dynamic and existing informal or formal power relationships. Interviews were tape-recorded and professionally transcribed verbatim. The quotes used to support the analysis in this paper were translated from Swedish to English by a professional translator after the analysis.

## Data analysis

A qualitative method was used, as this was deemed suitable to explore the staff's experiences of the sound environment and to identify their views on context-specific factors of importance for a subsequent intervention study. More specifically, inductive thematic analysis was used according to the method described by Braun and Clarke [21]. Thematic analysis is theoretically flexible, which means that it can be applied using different ontological and epistemological approaches. In this study we adopted a realist approach, as we aimed, and considered it possible, to explore the participants' experiences of their reality via interviews [21]. We chose an inductive, data-driven approach that would enable us to identify patterns and unexpected themes, and that would provide a rich, detailed, and multifaceted account of the underlying data, which was not preconceived by us as researchers. However, we adopted a reflexive attitude and used critical discussions to attend to the context and consider our pre-existing knowledge. It was assumed that the investigation of the sound environment and factors important for an intervention could result in a number of different themes rather than being explained by one single phenomenon. Hence, we aimed to present a description of the entire dataset relating to the research topic. This is appropriate when there is less knowledge and previous research on the topic, as described by Braun and Clarke [21]. This was also one reason why the inductive thematic analysis was deemed more suitable than, for example, grounded theory.

The initial coding and analysis was done by the first author (KG) according to the six phases described by Braun and Clark [21]. The first and second steps were to familiarize ourselves with the data by reading each transcript several times, while registering initial codes that captured interesting features of the data. The entire dataset was systematically coded. In a third step, emerging conceptual themes were identified. Themes were identified throughout based on their "keyness", described as capturing something important in relation to the research question [21]. Next, the list of main themes was reviewed and refined until a list of clearly defined main themes and sub-themes was established, capturing coherent data to create mutually exclusive themes. In order to strengthen trustworthiness and inter-rater reliability, two of the co-authors (SF and SW) also read all the interviews and checked the coding done by the first author. All co-authors discussed and revised the themes and reviewed the extracts until a final list of main themes and sub-themes was agreed on. Finally, the themes were named and defined, and specific quotes from the interviews were selected to capture and illustrate the essence of each theme. See Table 2 for an example of the analytical process.

## Ethics

This study has been approved by the Regional Ethical Review Board, Gothenburg, Sweden (Dnr 659–18). Permission to conduct the study was obtained by the management at all participating workplaces. Confidentiality was assured at the start of the interviews by clarifying that no names or identifying information would be published. This information was also included in the written informed consent form signed by the participants.

## Results

The analysis of the data resulted in identification of four main themes and a number of sub-themes (see Table 3).

### 1. A challenging and harmful sound environment

The sound environment was viewed as challenging at all four workplaces, with reports of various disturbing sounds. These sounds were to some extent specific to the individual workplace. Some of the disturbing sounds were viewed as unnecessary, while others were meaningful as they contained useful information. A few of the disturbing sounds could fit within both categories depending on the circumstances, for example children screaming. Examples of disturbing sounds are presented in Table 4.

**1.1 Unnecessary and disturbing sounds.** The sounds that were viewed as unnecessary and disturbing drew the staff's attention away from their work duties.

*What has been added lately, which I think is extra disturbing, is TV, radio, well foremost TV... I think it is really good that we have TV, we need to have possibilities to entertain the patient. I insist that the patient should use headphones, I can't concentrate on my work if not, I just can't.* (Intensive care)

**Table 2. An example of the process of abstraction.**

| Unit of analysis | Code | Sub-theme | Main theme |
|---|---|---|---|
| *"I feel I've become much, much more sensitive to sounds in general. Things I used to be able to handle are painful now. I almost have to leave the room, otherwise I feel ill . . ."* (Preschool) | Has become more sensitive to sounds and experiences sounds as painful; sounds make her feel ill. | The sound environment is causing hearing-related symptoms | Health-related effects of a challenging and harmful sound environment |

**Table 3. Main themes and sub-themes.**

| Main themes | Sub-themes |
|---|---|
| **1. A challenging and harmful sound environment** | 1.1 Unnecessary and disturbing sounds |
| | 1.2 Meaningful, but disturbing sounds |
| **2. Health-related effects of a challenging and harmful sound environment** | 2.1 The sound environment is causing hearing-related symptoms |
| | 2.2 The noise is causing stress |
| **3. A good sound environment is not prioritized** | 3.1 Demanding psychosocial working conditions |
| | 3.2 Budget constraints |
| | 3.3 Noise is not part of the systematic work environment management |
| | 3.4 Lack of peace and quiet |
| **4. Resourceful and motivated staff** | 4.1 Attempts to handle the challenging sound environment |
| | 4.2 Individual motivation for change |
| | 4.3 Organizational support needed |

**Table 4. Examples of unnecessary and disturbing sounds, and of meaningful but disturbing sounds at the workplaces.**

| | Preschool | Intensive care unit | Obstetrics ward |
|---|---|---|---|
| **Examples of unnecessary and disturbing sounds** | Toys rubbing against specific materials<br>Noise in the dining hall with many children present, such as knives and forks on plates<br>Reconstruction work<br>Room acoustic conditions (e.g. rooms with an echo)<br>Screams from children without any apparent reason | Mechanical beds<br>Electric heat blanket<br>Fans and cooling systems<br>Patients watching TV or listening to music<br>Old trolleys (used for transportation of materials at the unit)<br>Doors opening and closing<br>Technical equipment (e.g. epidural anesthesia pumps)<br>Doorbell (when visitors are not allowed) | Equipment for administering laughing gas<br>Technical equipment (e.g. cardiotocography equipment)<br>Computers<br>Ventilation<br>Reconstruction work<br>Heating cabinets<br>Patients listening to music/radio<br>Room acoustic conditions (e.g. rooms with an echo) |
| **Examples of meaningful but disturbing sounds** | Screaming children (e.g. informing about the children's activities and mood)<br>Crying children (e.g. informing that a child needs comforting)<br>Children at play (e.g. informing about the children's interaction and communication skills) | Alarms<br>Doorbell (when visitors are allowed) | Women screaming while giving birth (e.g. informing about the birthing progress and condition of the mother)<br>Shift change (a lot of staff talking in a limited space) |

**1.2 Meaningful, but disturbing sounds.** Many sounds that could be described as loud or disturbing also provided important and meaningful information to the participants and guided them to some sort of action. For example, preschool teachers explained that they needed to be constantly attentive to sounds, in order to be aware of what was going on among the children. Having to be constantly alert and not being able to "turn off" the listening was viewed as demanding.

> *Because here, you need to be alert, so . . . you need to have your tentacles out there to, like, what's happening? You need to observe what is going on, what are they talking about? What is happening? What is that child doing? Oops, maybe someone is drawing there, and there something crashes, what was it? Was someone hurt? So, you need to check the situation.* (Preschool)

At the ICU, the loud and frequent noise from medical equipment provided important information regarding the patients. The participants felt that the use of alarms in health care had increased over the years.

*But the alarms are really good, and they are telling us something all the time and they have become . . ., it wasn't like that . . . When I started this line of work. . . it wasn't the same alarm stuff back then, that I can recall. So, it is not like you shouldn't have any alarms–you need to have that, absolutely.* (Intensive care)

## 2. Health-related effects of a challenging and harmful sound environment

The participants described that they believed that the work environment was the cause of several hearing-related symptoms. They also experienced that the noise in the workplace was causing stress.

### 2.1 The sound environment is causing hearing-related symptoms

Hearing-related symptoms, such as sound-induced auditory fatigue, sound sensitivity and tinnitus, were reported from all participating workplaces, and the participants believed that the work environment was the main cause of these symptoms. They described how certain situations at work with loud noises had caused hearing-related symptoms.

*Many of the midwives over the years have said that they're sure they got tinnitus at work because they can almost remember when it happened. They were in a room standing with some woman who screamed right into one of their ears and after that it was never good again.* (Obstetrics ward)

They also described how symptoms developed over time and could be felt after working extended hours such as during night shifts.

*I also think that I'm becoming more sound-sensitive . . . at least I feel that I'm becoming much more sound-sensitive during my spare time. I really suffer from . . . and, yes, it has developed over the years I think. When I've been working two nights then I feel I'm really sound-sensitive. If I go and do some exercise . . . when I do an exercise class I have to wear hearing protection.* (Intensive care)

**2.2 The noise is causing stress.**   Noise was perceived to be connected to stress. Noise at the workplace was causing stress, and stressful working conditions were making the noise more difficult to handle.

*I actually think a lot of noise causes inner stress. I also think it's something that gets everyone wound up, and it's such a big, fast-paced department . . . and that causes inner stress. I think a lot of people are worn out when they get home, not just because it was so busy, but because of that inner stress that gets everyone wound up. You can't deal with noise; you sit in your car on the way home, absolutely wrung out.* (Obstetrics ward)

The connection between noise and stress was also apparent in the experience of being highly aware of alarming sounds in general, even outside work. This can be described as tension and hypervigilance and a related stress reaction, which involved the participants automatically reacting to sounds that could signal that some form of action was needed.

*Then there is this with sound in general . . . So, like this, you're so used to that you're supposed to react if there is some sound. Like you know, someone is falling. You are there and it doesn't matter if I'm here or as a private person in the store. I hear something. And then you have this reaction like . . . What was that? You're almost on the way to, or when someone is screaming, you just . . .. You are like sort of tense. You are being alert all the time.* (Preschool)

### 3. A good sound environment is not prioritized

During the interviews, the participants described various obstacles to reducing disturbing and harmful noise at the workplace. Obstacles included challenging working conditions, a limited budget, a lack of systematic routines relating to noise and a lack of spaces for peace and quiet. It appeared that noise was not a prioritized area in the general work with health and safety at the workplace.

**3.1 Demanding psychosocial working conditions.** All four workplaces faced challenging psychosocial working conditions that had a negative impact on the sound environment, according to the participants. In preschool, the main problem the participants described was the large number of children in the groups. This was both stressful and noisy. The preschool staff strongly expressed that they wanted fewer children in the groups.

*It's like we've always said . . . or I have, at any rate, and I've been doing this for a hundred years . . . we know what it was like to have fewer kids and more staff . . . we want fewer children so that we have enough time for everything.* (Preschool)

At the obstetrics ward, there was a shortage of staff, a high workload and a lack of rooms and beds for patients. With all these issues, noise and the acoustic environment were not viewed as a priority.

*Things like this [noise] sort of get lost in the ruckus. We're trying to keep our heads above water and get enough people to cover every shift . . . So, the acoustic environment . . . I don't think anyone at the managerial level has the energy to deal with it.* (Obstetrics ward)

**3.2 Budget constraints.** The participants described that the constrained budget was one reason why there was a lack of focus on reducing the noise and improving the acoustic environment at the workplace.

*. . . even higher up in management, I doubt many of them have any idea of what we're subjected to every day . . . it's the budget that rules.* (Preschool)

**3.3 Noise is not part of the systematic work environment management.** There appeared to be a lack of coordination and routines regarding protecting staff members' hearing, and there was no routine for hearing protection wear. At the preschool there was even some confusion about whether it was allowed to wear hearing protection at work.

*I don't think we're allowed to wear them [earplugs].* (Preschool)

At the obstetrics ward, earplugs were used by some of the staff, but there was no routine regarding the provision or use of earplugs. At the ICU, some participants described that the question of noise was on the agenda and that they were trying to find solutions to improve the sound environment. Still, there appeared to be a lack of routines to protect the staff's hearing.

For example, some of the medical equipment was very loud, and one of the machines had hearing protection attached, indicating that this should be used when handling the machine. However, the participants had not received any information regarding whether and when the hearing protection should be used, and they were not aware of anyone using it.

*We have some hearing protection on the transport ventilator, because it's really loud when you adjust the oxygen . . . So, there is hearing protection attached . . . but I don't know if anyone uses it.* (Intensive care)

**3.4 Lack of peace and quiet.** There was a lack of quiet and peaceful places for taking breaks from work and from the noise at work.

*So, there are kids that stand there [outside the room] and almost knock on the window when you sit down to have your break.* (Preschool)

At the obstetrics ward there had been extreme situations at times when they had had a lot of patients and family members visiting them, which meant that it was difficult for the staff to find peace and quiet to complete their work tasks. This was clearly a very stressful situation for the staff.

*We've had situations when we're really busy, with too many patients. . . and you just can't get away. Somebody is bound to grab my arm and start talking the second I sit down to try and get something done . . . it's not loud noise, really, but there's nowhere you can go for a little peace and quiet.* (Obstetrics ward)

## 4. Resourceful and motivated staff

Participants from all four workplaces expressed motivation to make changes to improve the sound environment. Several solutions were proposed by the participants, including both adjustments to the physical environment and ways to organize work differently.

**4.1 Attempts to handle the challenging sound environment.** The participants explained that they were already actively creative in finding their own strategies to handle loud and disturbing sounds. It could be said that they did the best they could to deal with the noise using the available resources. The attempts that they described included using more sound-absorbing materials and reorganizing work-related tasks.

*Instead of trying to get help . . . we take a carpet and we . . . then we work by ourselves to make the best of it. . . . we are pretty much used to, you know, fixing things ourselves.* (Preschool)

At the intensive care ward, solutions to handle alarms from technical equipment were to turn off the alarm or lower the volume on the alarm signal.

**4.2 Individual motivation for change.** As described under the previous sub-theme, the participants were already creative in finding different ways to handle the sound environment. However, during the interviews it became apparent that the participants were motivated to make further changes. Increasing accessibility to hearing protection and reminding the staff to use it in situations with loud noise was one example where participants were motivated to put in more effort.

*We have to bring that up again at the workplace meeting and make hearing protection available . . . order them [more sets] . . . and to make them visible again, you know, to talk about it.* (Obstetrics ward)

Improving the sound environment for the children's or the patients' sake was perceived as a strong motivator for change. Changes in routines, or in how to organize work in order to improve the sound environment for the staff, were perceived as having a positive impact also on the children and patients in that environment.

*But you think, like, if we do stuff that is good for us adults then it will be beneficial to the children as well . . .* (Preschool)

**4.3 Organizational support needed.** Despite the fact that the participants were resourceful and had created different solutions to handle the noise, there were several changes that could only be implemented with support from the organization. These would require economic resources and/or structural changes. For example, for preschool, the solution that appeared to be the most appealing to the participants was to have fewer children in each group.

*To reduce the number of children in the groups. That is the most concrete [action]. Because there is a huge difference when there is, like, some bug going round . . . When seven, eight children are absent on the same day. You're like . . ., "How nice!"–It's terrible . . .* (Preschool).

Another example was creating quiet areas where it would be possible to get rest and recover from a noisy and stressful situation.

*So, one thing that I was thinking about, that wouldn't be so bad, is if we had a quiet room [the participant had a specific location in mind]. You know, those who liked that, could just go up and sit with their phone or . . . because there you can just put up your feet and . . . there are large windows, so you can just sit there and look at the trees . . . You know, it's right outside the dressing room but there are no alarms and there are no bells. It's quite nice to sit there actually.* (Intensive care)

Further examples included changing the signal of the doorbell so that it was only heard by certain members of staff, installing sound-absorbent panels, and constructing separate soundproof cubicles where personnel could do cognitively demanding work. Some suggested solutions involved very little cost and would be relatively easy to implement. At the maternity ward, suggestions for change included wearing hearing protection during the critical phases of childbirth, playing relaxing music at the entrance and dimming the lighting in the corridor to create a calm atmosphere.

*I think that if we could have . . . this, like, . . . there is this relaxation music . . . if we have that already when one goes through the entrance, then you get a different feeling in your body . . . And it doesn't cost anything.* (Obstetrics ward)

## Discussion

The current study aimed to explore how workers in communication-intense workplaces in health care and preschools experience the sound environment, and further to identify factors of importance for a subsequent intervention study. Four main themes emerged from the thematic analysis. Below, each theme will be discussed in turn.

### A challenging and harmful sound environment

The first main theme, *A challenging and harmful sound environment*, highlighted that disturbing sounds can either be perceived as unnecessary, or as meaningful. The experience of noise

is to some degree subjective and contextual. For example, the common non-auditory health effect "noise annoyance" can be moderated by factors such as self-rated necessity of the noise and control, where unnecessary noise is often perceived to be more annoying and a lack of control over the source can increase disturbance [22]. In addition, the meaning and predictability of the sounds can influence reactions such as the stress response [23]. There are also studies showing an evaluative aspect and that the attitude towards the sound source may affect annoyance [24]. Moreover, the type of work activity can affect the annoyance and disturbance response, for instance when noise masks important acoustic information, when irrelevant speech disturbs concentration when reading, or when noise is particularly disturbing during more cognitively complex tasks [22]. Much like several other preschool studies have shown [7], this study found that screaming from children was perceived by the staff as disturbing. We found that the context and information content could also influence whether screaming was perceived as unnecessary or meaningful. Similarly, previous research conducted in ICUs found that the same sound can be experienced by patients as disturbing on one occasion and comforting on another [19]. Given these results, an intervention aimed at improving the sound environment should focus on identifying which sounds are disturbing and particularly the contexts in which certain sounds are perceived as unnecessary, and then target these.

## Health-related effects of a challenging and harmful sound environment

The second main theme was *Health-related effects of a challenging and harmful sound environment*. Tinnitus, sound-induced auditory fatigue, and sound sensitivity were perceived as common symptoms, and the participants described that these symptoms were caused by the work situation. Regarding previous research on health effects of occupational noise exposure, noise-induced hearing loss and tinnitus may affect workers in many sectors, from construction to the social services to preschools [2, 25–27]. Of high relevance for the current study, our previous research among preschool teachers and obstetrics personnel showed an increased risk of several hearing-related symptoms, such as difficulty perceiving speech, tinnitus, hyperacusis and a symptom we termed "sound-induced auditory fatigue" [8, 10, 28]. Sound levels in preschool and obstetrics care have been found to reach or exceed the lower exposure action value and the noise exposure limit of 80 and 85 dB time-average A-weighted noise level for a nominal 8-hour working day (LEX,8h), established by the Swedish Work Environment Authorities as they pose a risk of damage to hearing [7, 9, 10, 29]. One study of preschool teachers reported that no subjects were classified as having hearing damage, defined according to the authors as mean pure tone hearing thresholds >35 dB HL at 2 and 3 kHz and >45 dB HL at 4 and 6 kHz, although the mean hearing thresholds for the study group were higher than the 50th percentile of an age-matched reference population [7]. Another study, reported that results from pure tone audiometry tests and distortion product otoacoustic emission (DPOAE) test were correlated with a calculated cumulative occupational noise dose among obstetrics personnel, such that an increase in noise showed significantly higher hearing thresholds at 6 kHz and 8 kHz bilaterally and at 3 kHz and 4 kHz in the left ear, and decreased DPOAE amplitude averaged over the frequency range 3 to 6 kHz and the 3 to 10 kHz range bilaterally [29]. Although the current study cannot make claims regarding causal effects, the staff clearly expressed that they perceived effects on hearing caused by the sound environment at work.

In addition to auditory effects, studies have confirmed noise to be associated with health outcomes such as long-term stress, annoyance, sleep disturbance, reduced cognitive performance, and cardiovascular diseases [30, 31]. Noise can be described as a stressor with acute activation of the stress axis that may, if prolonged or repeated, result in chronic health effects, as outlined by Babisch [3], Münzel, Schmidt et al. [31]. The stress load for both the personnel

and the children or patients may be related to the sound environment. Whereas a pilot study failed to reduce staff burnout by implementing use of hearing protection devices in preschools and schools, the intervention group did not increase in burnout while the control group did [15]. Among ICU personnel, non-auditory stress-related symptoms such as irritation, fatigue, tension headaches, and difficulties concentrating have been found to be prevalent [13].

In the current study it was also reported that the sound environment was causing stress symptoms. The lack of quiet breaks reported by the participants in our study may have implications for future health outcomes by long-term activation of the sympathetic nervous system. The participants in this study clearly expressed a need for peace and quiet to decrease the health effects of both noise and psychosocial stress.

## A good sound environment is not prioritized

The third main theme was *A good sound environment is not prioritized*. The participants from all workplaces included in the current study described demanding psychosocial working conditions and the psychosocial working conditions were perceived to have a negative impact on the sound environment. In spite of this, the described lack of routines to handle noise issues as an occupational hazard and the lack of routines for hearing protection highlighted that noise was not highly prioritized in the participating organizations. The high workload often found in human service occupations may explain why the sound environment has not been prioritized: either because other issues such as the psychosocial environment are more pressing, or because there is simply not enough time and energy to prioritize the sound environment. The non-prioritization of occupational noise in communication-intense workplaces could also be viewed in a larger societal perspective. For example, communication-intense workplaces in education and health care tend to be female-dominated. There is less research investigating noise in female-dominated workplaces than in more male-dominated, industrial workplaces [32]. The lack of previous research has limited the interest and knowledge for the hazardous effects on noise within communication-intense workplaces. In addition, it appears that noise rarely is part of the systematic work environment management. The regulations regarding occupational noise exposure in Sweden [33] mainly focus on attenuating the source of the noise. However, limiting/attenuating the sound from the source is not directly applicable on communication-intense workplaces where children or patients often are the noise source. Taken together, the lack of time, knowledge and appropriate tools for mitigating noise from human activities seem to be a hinder for prioritization.

## Resourceful and motivated staff

The last main theme identified was *Resourceful and motivated staff*. The staff had already made different attempts at dealing with the disturbing and demanding sounds themselves, often with limited resources and at low cost. The results regarding motivation can be understood in the perspective of the theory of individual readiness for organizational change [34]. Readiness for change is influenced by the extent to which employees believe that they are capable of implementing organizational change, and that the proposed change is appropriate for the organization. In the current study the participants could be described as having high levels of individual readiness for organizational change as they expressed capability to implement change, and talked about the appropriateness and benefits of changes to improve the sound environment. They suggested several concrete solutions, such as creating quiet rooms, adapting doorbells, installing absorbents and increasing the availability of hearing protection.

There have been some previous intervention studies aiming to improve the acoustic environment in preschool and health care institutions. The results have been mixed. Some studies

found no significant effect of the interventions. For example, Sjödin et al. [16] investigated acoustic and organizational interventions that aimed to reduce noise levels in preschools. The results showed that neither the acoustic nor the organizational interventions had a statistically significant impact on the subjectively rated sound level. Positive effects of interventions were found in a study in an inpatient neuroscience unit at a hospital, which aimed to reduce noise levels for patients and staff [35]. It was found that the noise reduction strategies resulted in a more quiet work environment and the authors concluded that involving committed staff had been crucial to achieve the changes.

The second aim of the study was to identify factors to guide the tailored interventions that was going to be implemented. Once the analysis was completed, the results were fed back to three of the participating workplaces: the obstetrics ward, and the two preschools. The intensive care unit was unable to continue with the planned intervention study. The themes found in the analysis, representing important factors for improving the work environment, were presented in workshops with managers and staff at each workplace. The results were discussed and translated into tailored interventions for each workplace. The workshops and the intervention process, including the specific interventions chosen, have been published elsewhere [36].

## Strengths and limitations

It may be possible that the participants found it difficult to fully express their opinions in the focus groups. The interviewers informed the groups that there was no need for consensus in the group, that all opinions, experiences and ideas were welcome. Moreover, the managers were not included in the focus groups to avoid participants' tailoring their responses in the presence of superiors. The sample used included individuals who had volunteered to participate in the interviews. Thus, it is possible that other employees at the workplaces would have reported somewhat different experiences and ideas. The results of the study were fed back to three of the participating workplaces and the results made sense to them, which strengthens the validity of the results. Because of difficulties recruiting participants, the focus groups were smaller than planned. Nevertheless, active group discussions emerged in all focus groups. No males participated in the interviews. This was because there were very few or no males working at the participating workplaces. A further limitation was that four of the interviews were conducted by two of the authors (KG and SW) and three were conducted by only one of the authors (KG). It is possible that the number of interviewers influence the interviews, however, both authors were experienced in conducting focus group interviews. Strengths of the study included researcher triangulation. Several researchers were engaged in the analysis and interpretation of the data. This increased the trustworthiness of the findings [37]. In addition, when examining the data, we judged that the interview data did not suffer from any one participant dominating the discussions. Regarding transferability of the findings, it can be expected that the themes are relevant to other, comparable communication intense workplaces. In qualitative research, it is also important to relate the findings to previous research and thereby add to the accumulation of knowledge [38]. In this paper, the study's main themes have been discussed in relation to previous research.

## Conclusions

In conclusion, it was found that workers in communication-intense workplaces in health care and preschools experienced the sound environment as challenging and at times harmful, causing hearing-related symptoms. Despite this, the need for a good sound environment did unfortunately not appear to be a priority of the participating organizations. Nevertheless, the participants expressed motivation for change and ideas for solutions. The results

indicate that solutions to improve the sound environment need to be innovative and context specific, as the work in communication-intense environments requires a good ability to hear and communicate. For example, the use of hearing protection may prove challenging, as may reducing noise at the source. It seems important to increase opportunities for auditory rest at work, for example by having quiet spaces at the workplace. Introducing routines to inform staff about health risks related to noise and to offer hearing protection that can be used in particularly noisy situations are further suggestions. Constrained budgets, coupled with the fact that noise seem to be down-prioritized in these communication-intense workplaces, limit the possibility of change. However, the participants suggested changes that were not necessarily costly; therefore, a further implication of this research may be that opportunities should be facilitated for staff to find their own solutions to improve the sound environment.

## Author Contributions

**Conceptualization:** Kristina Gyllensten, Sofie Fredriksson, Stephen Widen, Kerstin Persson Waye.

**Funding acquisition:** Kristina Gyllensten, Sofie Fredriksson, Stephen Widen, Kerstin Persson Waye.

**Investigation:** Kristina Gyllensten.

**Methodology:** Kristina Gyllensten, Sofie Fredriksson, Stephen Widen, Kerstin Persson Waye.

**Project administration:** Kristina Gyllensten, Sofie Fredriksson, Stephen Widen, Kerstin Persson Waye.

**Supervision:** Kerstin Persson Waye.

**Writing – original draft:** Kristina Gyllensten, Sofie Fredriksson, Stephen Widen, Kerstin Persson Waye.

**Writing – review & editing:** Kristina Gyllensten, Sofie Fredriksson, Stephen Widen, Kerstin Persson Waye.

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
