## [Decision Letter · Decision Letter 0]

9 Nov 2022

PONE-D-22-27631The experience of noise in communication-intense workplaces: a qualitative studyPLOS ONE

Dear Dr. Gyllensten,

Thank you for submitting your manuscript to PLOS ONE. After careful consideration, we feel that it has merit but does not fully meet PLOS ONE’s publication criteria as it currently stands. Therefore, we invite you to submit a revised version of the manuscript that addresses the points raised during the review process.

We look forward to receiving your revised manuscript.

Kind regards,

Mohammad Hossein Ebrahimi

Academic Editor

PLOS ONE

Journal Requirements:

"The study was funded by Forte (2016-07193)"

"- The study was funded by Forte 

 - Grant number: 2016-07193

 - KPW, SF, SW and KG received the award funded by Forte

 - https://forte.se/

Additional Editor Comments (if provided):

Reviewers' comments:

Reviewer's Responses to Questions

**Comments to the Author**

1. Is the manuscript technically sound, and do the data support the conclusions?

Reviewer #1: Yes

Reviewer #2: Yes

Reviewer #3: Yes

Reviewer #4: Yes

2. Has the statistical analysis been performed appropriately and rigorously? 

Reviewer #1: N/A

Reviewer #2: N/A

Reviewer #3: N/A

Reviewer #4: Yes

3. Have the authors made all data underlying the findings in their manuscript fully available?

Reviewer #1: Yes

Reviewer #2: Yes

Reviewer #3: No

Reviewer #4: Yes

4. Is the manuscript presented in an intelligible fashion and written in standard English?

Reviewer #1: Yes

Reviewer #2: Yes

Reviewer #3: Yes

Reviewer #4: Yes

5. Review Comments to the Author

Reviewer #1: A small editorial comment for the proof reader: Line 100 need to change the word 'of' to 'on' (cf. line 395)

Despite a small sample size (16 participants among four sites) inherent with much bias ie. selected study sites, volunteer participants, some participants were from a site where a noise study had previously been conducted, different interviewers with potentially different methods of interviewing) the study is relevant and the outcomes actionable. The study protocol is not complex and could be replicated in other settings with modifications. The thematic analysis approach focused the results into areas of information applicable to actual situations. The results were largely based on offerings from front line workers and less on interviews with managers who could provide solutions from an administration perspective The paper did not provide examples of interview questions or if they differed between focus groups, ie. front line versus managers. The paper includes an extensive citation of related, fairly recent publications for further exploration on this topic.

Reviewer #2: Reviewer comments

The authors present information on an interesting topic

Below are a few comments for consideration

Methods

• Table 1 is unclear. Were participants interviewed more than once?

• How many people were in each focus group discussion?

• Were managers also interviewed with focus group discussions?

• In places were there was only one manager what approach was used in data collection?

Ethics

• Authors should provide a more detailed ethical consideration information. Was permission sort from facility management? Was permission obtained from any other body before entry into the facility granted? How was confidentiality assured?

Discussion

• The second part of the aim is not clearly addressed in the discussion. What do all these findings mean? What factors did authors identify to guide tailored interventions in these settings?

Strengths and limitations

• Though state as a limitation, why was focus group discussion approach used instead of in-depth interviews given that the numbers involved in each site were very few?

• How did authors ensure that the presence of other work colleagues didn’t influence their opinion?

• How was triangulation of findings done?

Reviewer #3: This is a good piece of research that I enjoyed reading. It is well written and represents an original research.

Nevertheless, I have just a few comments on the work.

1. Table 1: give a definition for KG and for SW, as a footnote beneath the table. A table should be self-explanatory without referral to the text to understand its content.

2. The sample used was a convenience sample, where only people who volunteered to participate were included. This introduces selection bias to the study. This should be addressed as a limitation of the current study.

3. In the data collection process, several researchers participated in interviewing participants. This can easily introduce interviewer bias into the study. Since different information can be elicited from different participants depending on the differences between interviewers in the way they perform the interview. This has been mentioned by the author as a strength in the discussion, but I see this as a limitation.

Although data have not been made fully available, this has been explained by the author with convincing rationale.

Reviewer #4: Thank you for the opportunity to review this article which is an interesting manuscript.

The theme of this article is very important.

The title is clear, and the abstract covers the main aspect of the

The authors may need to provide some additional information on how participants were recruited and how they decided a number of participants were sufficient for the statistical analysis in this study. What sampling strategy did they use? How did they manage to obtain a representative sample in the end? The authors may need to give more information about the content of the interviews. They did not tell the readers what questions they asked in interviews

6. PLOS authors have the option to publish the peer review history of their article (what does this mean?). If published, this will include your full peer review and any attached files.

Reviewer #1: **Yes: **Helen Bangura, MHSc

Reviewer #2: No

Reviewer #3: No

Reviewer #4: No

---

## [Author Response · Author response to Decision Letter 0]

20 Dec 2022

Response to Reviewers 

Dear editor. We would like to thank you and the reviewers for the constructive and helpful comments. We have revised the manuscript accordingly. 

1. The manuscript has been revised in accordance with the style requirements.

2. The Funding information and Financial Disclosure have been revised so they match 

3. The funding information should read: 

- The study was funded by Forte 

- Grant number: 2016-07193

- KPW, SF, SW and KG received the research grant funded by Forte The Swedish Research Council for Health, Working Life and Welfare

 - https://forte.se/en

4. I have changed the affiliation to University of Gothenburg. 

Kristina Gyllensten, Department of Occupational and Environmental Medicine, University of Gothenburg, and Sahlgrenska University Hospital, Gothenburg, Sweden 

5. Data can only be shared on request as it contains sensitive information. We suggest that the following statement is included: 

Data cannot be shared publicly because potentially attributable sensitive information about health and symptoms regarding the participants. And when sharing such data there has to be an approval from a Swedish Ethical committee. However, anonymised data is available with an approval from an ethical review board. For data requests, contact: Department of occupational and environmental medicine, Gothenburg University, amm@amm.gu.se or Kristina Gyllensten, Department of occupational and environmental medicine, Gothenburg University, kristina.gyllensten@amm. gu.se. The name of the data set is ‘FriskArb: Noise in female-dominated occupations, the qualitative study.’

6. The reference list has been updated. 

Comments from the reviewers 

Reviewer #1: 

Comment: A small editorial comment for the proof reader: Line 100 need to change the word 'of' to 'on' (cf. line 395)

Reply: This line has been changed

Comment: The paper did not provide examples of interview questions or if they differed between focus groups, ie. front line versus managers.

Reply: Some of the interview questions have been included on page 7. 

Information regarding the fact that the questions did not differ between the staff and the managers has been included on page 7. 

Reviewer #2

Methods

• Table 1 is unclear. Were participants interviewed more than once?

• How many people were in each focus group discussion?

• Were managers also interviewed with focus group discussions?

• In places were there was only one manager what approach was used in data collection?

Reply: The table on page 6 has been revised and now includes information regarding size of each focus group and information about how managers were interviewed. 

Ethics

• Authors should provide a more detailed ethical consideration information. Was permission sort from facility management? Was permission obtained from any other body before entry into the facility granted? How was confidentiality assured?

Reply: Information about consent was added in the ethics section on page 9 

Discussion

• The second part of the aim is not clearly addressed in the discussion. What do all these findings mean? What factors did authors identify to guide tailored interventions in these settings?

Reply: The findings in relation to the second aim is discussed in the discussion section on page on page 25-26. 

Strengths and limitations

• Though state as a limitation, why was focus group discussion approach used instead of in-depth interviews given that the numbers involved in each site were very few?

Reply: Focus groups were used as this is a method that can promote discussions about different experiences and opinions among the participants. This is stated in the methods section on page 7. However, because of difficulties recruiting the groups were smaller than planned. This has been added as a limitation on page 25. 

Comment: How did authors ensure that the presence of other work colleagues didn’t influence their opinion?

Reply: At the start of each focus group the interviewers clearly stated that there was no need for consensus in the group and that everyone’s opinions and experiences were important. The interviewers also ensured that everyone got the opportunity to speak and paid close attention to the group dynamic as stated on page 7. However, it is not possible to fully ensure that colleagues didn’t influence each other. 

Comment: How was triangulation of findings done?

Reply: One of the authors was the main person responsible for the analysis, but two of the other authors read all interviews and gave their input to the analysis during several stages of the analysis. The list of themes and sub-themes were discussed and revised by all authors until a final list of themes was agreed on. This is described in the methods section on page 8-9. 

Reviewer #3

Comment: Table 1: give a definition for KG and for SW, as a footnote beneath the table. A table should be self-explanatory without referral to the text to understand its content.

Reply: Good suggestion, we have added a footnote under the table on page 7. 

Comment: The sample used was a convenience sample, where only people who volunteered to participate were included. This introduces selection bias to the study. This should be addressed as a limitation of the current study.

Reply: This is now addressed under the strengths and limitations on page 25. 

Comment: In the data collection process, several researchers participated in interviewing participants. This can easily introduce interviewer bias into the study. Since different information can be elicited from different participants depending on the differences between interviewers in the way they perform the interview. This has been mentioned by the author as a strength in the discussion, but I see this as a limitation.

Reply: Good point, we agree that data collection by different individuals can be a limitation. One of the authors were conducting all the interviews, and one of the authors participated in four of them. This was not clearly explained in the text, so we have revised the text under the strengths and limitations section on page 25-26 to make this more clear. We have also removed the sentence that described this as a strength on page 26. 

Reviewer #4: 

Comment: The authors may need to provide some additional information on how participants were recruited and how they decided a number of participants were sufficient for the statistical analysis in this study. What sampling strategy did they use? How did they manage to obtain a representative sample in the end? 

Reply: 

- A purposeful sampling strategy was used, meaning that we aimed to recruit participants with relevant experience regarding communication-intense workplaces and noise at work in health care or preschool. The participants were recruited via their managers who invited all relevant staff to participate. Information about this has been added in the methods section on page 6. 

- In qualitative research it is not possible to guarantee a representative sample, and each reader judges the transferability of the results. The transferability is discussed on page 26. 

- There was no statistical analysis performed in the study as this was a purely qualitative study. The number of interviews (7) and the number of participants (16) were considered to be sufficient to explore the research question. In qualitative research the number of participants and groups depends on the aim of the project. Kitzinger, often cited regarding focus groups, writes that focus group studies can consist of a varying number of groups, but that most studies involve just a few groups. 

Reference: Kitzinger, J. (1995). Introducing focus groups. BMJ, 311, 299-302. 

Comment: The authors may need to give more information about the content of the interviews. They did not tell the readers what questions they asked in interviews

Reply: Some of the interview questions have been included in the method section on page 7.

---

## [Editor Report · Decision Letter 1]

26 Dec 2022

The experience of noise in communication-intense workplaces: a qualitative study

PONE-D-22-27631R1

Dear Dr. Gyllensten,

We’re pleased to inform you that your manuscript has been judged scientifically suitable for publication and will be formally accepted for publication once it meets all outstanding technical requirements.

Kind regards,

Mohammad Hossein Ebrahimi

Academic Editor

PLOS ONE
---

## [Editor Report · Acceptance letter]

2 Jan 2023

PONE-D-22-27631R1 

The experience of noise in communication-intense workplaces: a qualitative study 

Dear Dr. Gyllensten:

I'm pleased to inform you that your manuscript has been deemed suitable for publication in PLOS ONE. Congratulations! Your manuscript is now with our production department. 

Kind regards, 

on behalf of

Dr. Mohammad Hossein Ebrahimi 

Academic Editor

PLOS ONE